# Research Progress on the Preparation and Applications of Laser-Induced Graphene Technology

**DOI:** 10.3390/nano12142336

**Published:** 2022-07-07

**Authors:** Yani Guo, Cheng Zhang, Ye Chen, Zhengwei Nie

**Affiliations:** 1School of Mechanical and Power Engineering, Nanjing Tech University, Nanjing 211816, China; guo17793470078@163.com (Y.G.); chenye@njtech.edu.cn (Y.C.); 2College of Engineering, Nanjing Agricultural University, Nanjing 210031, China; zhangcheng@njau.edu.cn

**Keywords:** laser-induced graphene, preparation process, signal sensing, environmental protection, energy storage

## Abstract

Graphene has been regarded as a potential application material in the field of new energy conversion and storage because of its unique two-dimensional structure and excellent physical and chemical properties. However, traditional graphene preparation methods are complicated in-process and difficult to form patterned structures. In recent years, laser-induced graphene (LIG) technology has received a large amount of attention from scholars and has a wide range of applications in supercapacitors, batteries, sensors, air filters, water treatment, etc. In this paper, we summarized a variety of preparation methods for graphene. The effects of laser processing parameters, laser type, precursor materials, and process atmosphere on the properties of the prepared LIG were reviewed. Then, two strategies for large-scale production of LIG were briefly described. We also discussed the wide applications of LIG in the fields of signal sensing, environmental protection, and energy storage. Finally, we briefly outlined the future trends of this research direction.

## 1. Introduction

Energy and environmental issues have gradually become a formidable challenge for human sustainable development. With the continuous deepening urgency of carbon peaking and carbon neutrality targets, all kinds of clean-energy and renewable-energy materials are ushering in unprecedented new opportunities for development. Graphene, a type of two-dimensional carbon nanomaterial with sp^2^ hybrid connected carbon atoms stacked into a honeycomb lattice structure, is considered as a revolutionary material of the future. Since graphene was discovered in 2004, it has attracted the attention of scholars because of its excellent physical and chemical properties, such as ultra-high carrier mobility (>250,000 cm^2^/V·s) [1], high light transmission (white light absorption up to 2.3%) [2], high thermal conductivity (3000~5000 W/m·K) [3,4], large specific surface area (2630 m^2^/g) [5], ultra-thin thickness (0.34 nm) [6], good mechanical properties [7], and chemical stability [8], etc. 

The traditional methods for preparing graphene mainly include chemical vapor deposition, redox, mechanical stripping, and epitaxial growth, etc. Chemical vapor deposition [9] is a process in which steam or gaseous substances react at a gas-solid interface to form solid deposits, and graphene materials can be grown on the surface of metal substrates. Kataria et al. [10] studied the method of growing graphene on different metal substrates, especially on Cu substrates, based on existing chemical vapor deposition technology. The quality of graphene achieved by this method was good, however, the synthesis route was long and costly. In addition, problems (grain boundary defects and uneven thickness) were prone to occur during the growth process. Redox graphene [11] was first obtained from graphite oxide by the Hummers method [12], then transformed into graphene oxide (GO) by ultrasonic dispersion, and finally, the oxygen-containing functional groups of graphene oxide were removed by various reduction methods (chemical reduction [13,14], thermal reduction [15], and microwave reduction [16]) to obtain graphene-like structures with excellent electrochemical and photochemical properties. This method is a simple preparation process, but the prepared graphene has certain defects. Mechanical exfoliation [17] is a method of repeatedly taping the surface of graphite materials with adhesive tape to exfoliate single or few layers of graphene. Although high-quality graphene can be obtained by this method, the size of graphene is generally only in the micron range [18] and the production efficiency is low, which cannot meet industrial production requirements. The epitaxial growth method [19] uses SiC as a feedstock to obtain large-scale graphene by thermal decomposition of Si on specific crystalline planes of 4H or 6H-SiC. The graphene obtained by this method has excellent electronic properties. The high-temperature environment required by this process limits the type of substrate that can be used to grow graphene directly, so additional transfer processes are usually required to grow graphene on other substrate materials, making the preparation cost higher [19,20,21].

These above methods for preparing graphene have complex processes, low efficiency, limited choice of substrate materials, and require additional patterning processes to achieve graphene patterning. Therefore, it is necessary to develop a technology with simple operation processes, low cost, high efficiency, wide choice of raw materials, controllable graphene patterning, and a more environmentally friendly preparation process.

In recent years, with the rapid development of ultrafast optics, fiber lasers and high-power lasers, laser technology has widely been used in the fields of materials processing, biomedical, and defense industries. Laser processing technology, as an ultra-fine, non-contact, maskless, and programmed processing method, embodies the advantages of high efficiency, flexible patterning, and precise physical property regulation in the preparation and processing of graphene. GO can be converted into reduced graphene oxide (RGO) with the help of laser reduction [22,23]. Compared with the conventional GO reduction method, the laser reduction method can remove oxygen-containing functional groups directly, without special environments (e.g., high temperature or vacuum) or toxic reducing agents, and the process is simple and easy to operate [24]. The laser irradiation method resulted in a 100-fold increase in the electrical conductivity of the originally insulating GO [23]. Although the electrical conductivity of RGO could be partially restored after laser treatment, the physical properties of the sp^2^ carbon lattice were changed due to a large number of defects. The RGO was not completely equivalent to graphene. Therefore, there are limitations in the preparation and application of this method in graphene-based electronic devices [22].

In 2014, James M. Tour’s team [25] successfully prepared porous three-dimensional graphene, also known as laser-induced graphene (LIG), using a CO_2_ infrared laser to directly scan on commercial polyimide (PI) films, as shown in Figure 1. Since laser irradiation generates a localized high temperature (>2500 °C) in the irradiated region, it breaks the C–O, C=O, and N–C bonds, causing the carbon atoms to rearrange and form a graphene structure. Meanwhile, the remaining carbon atoms recombine and release as gases. Due to the short conversion time, the carbon rings cannot rearrange into a completely regular 2D lattice structure, which is why the 5-, 6-, and 7-membered carbon rings are produced [26,27,28]. As demonstrated in Figure 1e, the appearance of the typical 2D peak in the Raman spectrum indicates that the black material on the surface of the PI is graphene [25]. 

Compared with the traditional hexagonal honeycomb lattice of two-dimensional graphene, LIG contains a large number of crystalline structures consisting of two pentagons and one heptagon lattice. Due to these unique lattice structures, the graphene can be folded and warped [25]. The laser preparation process effectively avoids the disadvantages of high temperature conditions, long reaction cycles, and low yields of traditional CVD methods, and achieves regioselective growth and patterning design of graphene. In addition, the lattice of chemically exfoliated graphene is usually over-modified, which weakens the performance of its associated devices. Laser-induced doping averts unnecessary impurity residues and defects usually occurring in the chemical doping process [29,30,31].

LIG technology can control graphene’s micro morphology and greatly improve its light utilization and photothermal conversion efficiency [32]. The three-dimensional frame structure of LIG provides a porous interface with rich defects and a large accessible surface area. Thus, LIG not only exposes a large number of active sites, but also facilitates the transportation of substances. The whole laser processing can be carried out in air. The patterned structures can be directly prepared without any solvent, environmental conditions, masks, or raw material pretreatment. Therefore, laser-induced technology has become one of the main means of preparing graphene, offering the possibility of its wide application in various fields.

In this review, we focus on the preparation process, property control of graphene materials, and their applications in the fields of signal sensing, environmental protection, and energy storage, highlighting the latest research results, as illustrated in Figure 2. First, the conventional preparation process of graphene, and the laser processing preparation process, are introduced. Second, the effects of laser processing parameters, laser type, precursor materials, process atmosphere, and doping on graphene morphology and properties are overviewed. Then, two routines for large-scale production of LIG are briefly described. Next, the research progress of LIG in several related fields is summarized, including various sensors, air filters, antibacterial/antiviral surfaces, water treatment, micro-supercapacitors, and batteries. Finally, the prospects and challenges of LIG in building a green and healthy world are discussed.

## 2. Process Control of LIG

The surface morphology, as well as chemical, electrical, and mechanical properties of graphene, are related to the laser parameters, laser type, process atmosphere and doping. In this section, we will briefly introduce the influence of various process parameters on LIG morphology and properties during LIG preparation.

### 2.1. Laser Processing Parameters

It was found that the morphology pore size, and thickness of the graphene structure could effectively be tuned by adjusting the laser parameters (e.g., energy and pulse repetition rate) [41,42]. If only LIG fibers (LIGF) of a few hundred microns were desired, a strong absorption laser could be chosen. Otherwise, a weak absorption laser could be chosen for LIGF of ~1 mm thickness. Moreover, a larger laser energy increased the rate and amount of gas release, leading to larger pore sizes and higher porosity. Too high laser energy could destroy the porous structure of graphene, while too low laser energy was not sufficient to convert the polymer into graphene. As shown in Figure 3a–d, with the increase of laser energy, PI underwent the carbonization and graphitization process [43,44]. When the laser energy increased to 5.5 J/cm^2^, PI was converted into graphene [41].

It is worth noting that scanning speed, laser power, wavelength, and pulse width also play a key role in the structure formation of graphene and should be considered simultaneously in the graphene preparation process. Varying these parameters can significantly affect the electrical conductivity, composition, and morphology of LIG (Figure 3e) [45,46,47,48]. Reasonable adjustment of laser power can effectively control the ratio of each element (Figure 3f) [49]. Wang et al. [50] prepared LIG on the surface of PI films using a 40 W CO_2_ infrared laser (wavelength 10.6 μm) and a 5 W UV laser (wavelength 355 nm). The experimental results showed that under the same UV and CO_2_ laser flux, the ablation of PI films formed different morphologies. The surface morphologies with micron-sized and nanometer pores were found under the UV laser, while the sheet structure with micron-sized pores and few pores was produced under CO_2_ laser. The laser energy density could be controlled with the laser pulse width. Garland et al. [51] tested the laser pulse widths of 10, 20, 30, 40, and 50 ms, respectively (Figure 3g). SEM showed that the LIG structure produced by high pulse width was relatively smooth and flat, and the surface roughness was low (R_q_ = 14.4 μm). The LIG produced by low pulse width had a high specific surface area and randomly formed nodule structures (R_q_ = 30.9 μm). 

Selecting appropriate laser parameters (e.g., pulse duration and repetition frequency) can increase the electrical conductivity of laser carbonized structures on PI surfaces. This is because the laser increases the conductivity by reducing the level of defects within the orbit during graphitization. However, the substrate thickness limits a further increase in conductivity. Biswas et al. [52] also demonstrated this idea by means of simulations. In addition, a recent publication by Chen et al. [45] demonstrated that both the laser scan interval and the out-of-focus distance had a significant effect on the conductivity and wettability of fluorine-doped laser-induced graphene (F-LIG) films. In the single-factor analysis, the scan interval was 30–50 μm and the out-of-focus distance was 5–7 mm, within which the laser parameters were adjusted. The F-LIG then had good superhydrophobicity and conductivity. It is noteworthy that both the conductivity and hydrophobicity of the material rapidly decreased with the increase of each parameter, which is consistent with the previously published conductivity results [46]. Furthermore, ordered porous graphene structures could be obtained under laser irradiation at a scan speed of 160 mm/s and a repetition frequency of 20 kHz, which exhibited superhydrophobic behavior at contact angles greater than 150° [47]. These observations are consistent with the previously published results on the wettability of carbon-based materials [53]. In addition, the spatial resolution of LIG obtained by different lasers is also different. LIG spatial resolutions of 40 µm, 12 µm, and 10 µm were obtained with a 343 nm UV femtosecond laser, a 405 nm visible laser, and a 10.6 µm CO_2_ laser, respectively [54,55,56].

**Figure 3 nanomaterials-12-02336-f003:**
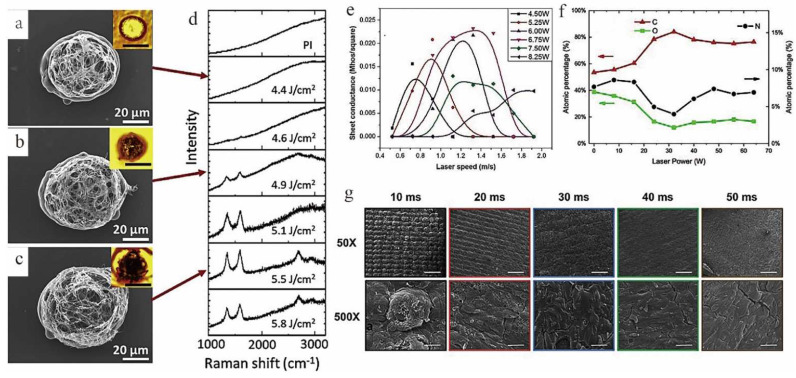
SEM images of samples with fluence set at (**a**) 4.4 J/cm^2^, (**b**) 4.9 J/cm^2^, and (**c**) 5.5 J/cm^2^; the insets in each SEM image are images of the same spot. (**d**) SEM images of the laser spot as fluence was increased (from top to bottom), adapted with permission from [41]. (**e**) Sheet conductivity of carbon trace on polyimide as a function of laser fabrication parameters (power and speed), adapted with permission from [48]. (**f**) The atomic percentage of C, N, O of PI and LIG fabricated with varied laser parameters, adapted with permission from [49]. (**g**) SEM micrographs for LIG created with distinct laser pulse widths, adapted with permission from [51].

### 2.2. Laser Type

There are three types of lasers used to prepare LIG: infrared CO_2_ laser, ultraviolet laser (UV), and visible laser.

Infrared laser preparation of graphene mainly stems from the photothermal effect, where the absorbed laser energy can be significantly converted into heat due to the high energy density of the laser, and the instantaneous high temperature can break and recombine the chemical bonds of the precursors [57], as shown in Figure 4a. PI films are the most popular precursor substrates for the preparation of LIG due to their robustness and extremely high thermal stability. Moreover, the presence of aromatic carbon in the chemical structure of PI makes it easier to form hexagonal graphene structures than other precursors. Many researchers have used infrared CO_2_ lasers (10.6 µm CO_2_ lasers typically produce line widths of 60–100 µm) for the irradiation of PI films to prepare LIG. Laser irradiation region generates localized high temperatures that lead to C–O, C=O, and N–C bond breakage and carbon atom rearrangement, resulting in porous graphene on the film surface [13,25,41,58,59,60,61,62,63,64,65,66,67,68].

For UV laser, the formation of graphene is mainly due to a photochemical reaction process. Due to the short wavelength and high energy of UV laser, especially when the photon energy is larger than the dissociation energy, the absorbed photon energy can directly break chemical bonds [26,57], as shown in Figure 4b. Together with the limitation of the Abbe diffraction limit, UV laser sources are thus good candidates for improving the spatial resolution of LIG devices [69]. Garland et al. [51] used a low-cost UV laser to produce LIG. The UV laser exhibited low absorption and photothermal heating within the PI substrate. The resulting carbon structures were non-homogeneous substrates of graphene or graphitic carbon, with less LIG than that of the structures produced by CO_2_ infrared lasers. 

For visible lasers, both photothermal and photochemical reactions may promote the formation of LIG [70]. Recently, researchers used a visible 405 nm wavelength laser to convert polymers into graphene with extremely small dimensions (spatial resolution of ~12 μm and thickness of <5 μm), which is close to LIG formed by UV lasers, as shown in Figure 4c. The characteristic size of LIG prepared with visible laser was reduced by at least 60%. It was almost 10 times smaller than the graphene obtained using infrared laser [56]. Therefore, this LIG can be directly used to make a flexible humidity sensor for sensing human breathing [56]. In addition, the 405 nm visible laser and the blue-violet laser were shown to form LIG from various carbon sources [71,72].

### 2.3. Precursor Materials

Many scholars have prepared LIG from various precursor materials, as shown in Table 1. The choice of precursor materials is also particularly important in LIG preparation. Researchers have conducted numerous exploratory studies for this purpose. In the early days of LIG discovery, only a few materials, such as PI and polyetherimide, could be successfully converted into graphene. Precursor materials that were later reported for LIG formation include, but are not limited to, carbon precursors containing lignin (pine, coconut shell, and potato peel), high-temperature engineering plastics (PI, Kevlar, PSU, PPSU, PES, PEI, PPS, SPEEK, and PEEK), thermosetting resins (PR and cross-linked polystyrene), and other materials (paper, silk, and xylan) [44,49,54,61,62,65,68,69,70,71,72].

For thermally sensitive materials containing high cellulose content, such as paper, wood, and fabrics, the addition of flame retardants to inhibit ablation and volatilization in ambient air promotes LIG formation [44,49,62]. Thus, two irradiation steps are required to form LIG: the first is defocusing, which converts cellulose into carbon, and the second is focusing (or near-focusing), which converts carbon into LIG [49]. It was proposed that using a UV femtosecond pulsed laser, instead of the conventional CO_2_ laser, could achieve a lower graphitization temperature due to the ultra-short pulse duration, which is shorter than the time required for heat transfer to the surrounding environment. The results showed that this method could effectively reduce the degree of ablation of wood, and could convert wood and leaves into graphene with good electrical conductivity in an air environment without protective gas [73].

Using laser-induced technology to extract high-quality graphene from nature's abundant precursor materials is simple, economical, and environmentally friendly, laying a solid foundation for the large-scale application of graphene.

**Table 1 nanomaterials-12-02336-t001:** Laser induced graphene process and applications.

Materials	Environment	Laser	Application	Refs.
PI	Ambient air	10.6 µm CO_2_ laser	Micro supercapacitor\ Sensor\ Air filter\ Seawater desalination	[25,35,49,74,75]
	Ambient air	405nm visible laser	Sensor	[70]
	Ambient air	405nm blue_violet laser	Micro supercapacitor	[56]
	O_2_/Air/Ar/H_2_/SF_6_/	10.6 µm CO_2_ laser	Micro supercapacitor\ Water treatment\ Air filter	[63]
Paper	Ambient air	10.6 µm CO_2_ laser	Sensor\ Micro supercapacitor	[46,58,76]
Cloth	Ambient air	10.6 µm CO_2_ laser	Micro supercapacitor	[58]
Food	Ambient air	10.6 µm CO_2_ laser	Micro supercapacitor	[58]
Xylan	Ambient air	10.6 µm CO_2_ laser	Sensor	[76]
Pinewood	N_2_	1064nm nanosecond and picosecond laser	-	[77]
Wood	Ambient air	UV laser	Sensor\ Supercapacitor	[55]
	Ar/H_2_	10.6 µm CO_2_ laser	Supercapacitor	[66]
Leaves	Ambient air	UV laser	Sensor\ Supercapacitor	[38,55]
PSU	Ambient air	10.6 µm CO_2_ laser	Water treatment\ Fuel battery	[78]
PEEK	Ambient air	10.6 µm CO_2_ laser	-	[79]
SPEEK	Ambient air	10.6 µm CO_2_ laser	Supercapacitor	[80]
PR	Ambient air	405nm visible laser	Sensor\ Supercapacitor	[70]
Silk	Ambient air	Fiber laser	Sensor	[81]

### 2.4. Doping and Process Atmosphere Control

Although the unique structure of graphene gives it excellent thermal, mechanical and electrical properties, the application of intrinsic graphene in electronics is limited by its zero-bandgap property. It is especially important to obtain graphene with a bandgap tunable within a certain range. To open the bandgap of graphene and solve the application question of graphene in electronics, researchers have explored many methods. One of the most feasible methods is to improve the performance of graphene, or obtain the properties that graphene does not have, by doping. This is because a carbon structure doped with heteroatoms can provide many active sites for electrochemical reactions, thus facilitating ion transfer during electrochemical processes [82,83].

From the research status of graphene doping, doping technology can be basically divided into in situ and non-in situ. In situ doping is usually achieved by changing the substrate composition or gas environment during laser ablation (see Table 2). Ye et al. [67] mixed N-methylpyrrolidone with metal complexes in liquid amidic acid and obtained PI mixed with metal complexes through a multi-step process. The metal complexes were thermally decomposed into metal oxide nanoparticles using laser irradiation, and then LIG-MoO_2_, LIG-Co_3_O_4_ and LIG-Fe_3_O_4_ were successfully prepared. The above-mentioned graphene had excellent redox catalytic activity and is expected to be applied in the electrocatalysis of fuel cells. Clerici et al. [84] and Chhetry et al. [85] deposited molybdenum disulfide on PI by spin-coating and hydrothermal methods, respectively, followed by laser treatment to obtain porous MoS_2_-LIG, which had good electrochemical properties and high surface area.

In addition, heteroatoms, such as B, N, P, and S, are also important dopants. The direct use of substrate materials containing these atoms or the indirect introduction of these atoms can effectively improve the electrochemical properties of graphene. Peng et al. [65] first prepared PI sheets containing boric acid using H_3_BO_3_ solution with PAA, followed by the preparation of B-LIG using a CO_2_ laser under ambient conditions, resulting in B-LIG being able to be placed into active electrodes that could be used in flexible micro-supercapacitors. Due to the boron doping, the maximum area capacitance of the prepared devices reached 16.5 mF/cm^2^, which was three times higher than the undoped devices, and the energy density increased by 5–10 times, making this boron-doped LIG material a great potential for future microelectronics applications. Zhang et al. [86] prepared high concentration (≈13 at%) nitrogen-doped PI containing urea in a nitrogen atmosphere graphene, and used it as an anode for sodium ion batteries, which exhibited excellent rate capability and good cycling stability. Later, Yang et al. [87] also prepared phosphorus-doped graphene by a similar method, which improved graphitization due to ammonium polyphosphate providing not only phosphorus but also flame retardancy. Singh et al. [78] prepared high-quality sulfur-containing porous graphene structures in three polysulfone polymers: PSU, PPSU, and PES. Due to the excellent electrochemical and antifouling properties of this sulfur-doped LIG, it can effectively be used in wastewater purification and antifouling cathodes in microbial fuel cells.

It is very noteworthy that multi-element co-doping is more effective than single element doping due to the synergistic effect of co-doped elements, which can form stronger active regions on its surface [29,88]. An LIG co-doped with N and B (NB-dLIG) was reported in a recently published article [29]. The LIG surface obtained by the first laser pyrolysis was then coated with another layer of polyamido acid/H_3_BO_3_, followed by another laser irradiation, resulting in the incorporation of N and B atoms into the graphene-like structure. Due to the synergistic effect of N and B, the capacitive properties of NB-dLIG were significantly improved. This method provides a new idea for the preparation of multiple heteroatom co-doped LIGs [29]. Another in situ doping method was achieved by changing the process atmosphere during laser processing, such as laser irradiation of PI in Ar, H_2_, O_2_, and SF_6_ atmospheres, which effectively modulated LIG surface superhydrophobicity or superhydrophilicity [63].

Non-in situ doping refers to the subsequent modification and coating treatment of the formed LIG. Various composites have been successfully prepared by electrodeposition of materials inside or on the surface of graphene. LIGs are commonly used with a variety of materials, such as commercial plastics (e.g., polyethylene), construction materials (e.g., latex paint, Portland cement, solid hydrocarbons, and epoxy resins), to form composites with different properties and applications [64]. Luong et al. [64] infiltrated a filler material into the porous LIG prepared by PI by gravity or hot pressing, and peeled off the PI layer, after it cured, to finally obtain LIG composites (LIGCs) with physical properties. This material had superhydrophobicity and antimicrobial contamination capability for antimicrobial applications and resistive memory device substrates. Li et al. [89] reported a laminated composite method in which LIG patterns were first laser-written on PI, followed by laminating the PI-supported LIG film with polymer, and peeling off unconverted PI layers to form large-area, multifunctional, robust, multilayered and patterned composites. The prepared composites can be applied in frictional electrical nanogenerators, biomedical films, and puncture detectors.

## 3. Scale-up Production of LIG

LIG technology has emerged as a powerful material fabrication method. Currently, two pathways exist for large-scale production of LIG: in-plane roll-to-roll method, and 3D printing of LIG on a macroscopic scale.

### 3.1. Roll-to-Roll Production of LIG

The roll-to-roll production process is feasible for continuous fabrication of LIG films, as shown in Figure 5a [90]. PI films are laser irradiated and converted into LIG under roller drive, followed by a catalytic cell, to obtain electrodeposited LIG. The PI films embedded with LIG are rolled up after drying. Wang et al. [91] proposed a new process for the preparation of graphene papers (GPs) by introducing a PI paper as the precursor medium. The PI paper had a special porous laminar fiber network structure that could uniformly absorb laser irradiation energy and had no strong shape distortion during the graphitization process, thus ensuring the processing of GP materials in large sizes. An LIG paper (LIGP) with an area of up to 1400 cm^2^ was prepared in their laboratory. This LIGP was not dependent on any substrate, and could be bent, folded, and cut into specific shapes, which gives it the characteristics of scalable size, mechanical flexibility, and customizable shape. In addition, the authors also indicated that the laboratory-scale process could be upgraded to roll-to-roll continuous fabrication in the future.

### 3.2. 3D Printing of LIG

Complexly-shaped graphene components with thicknesses in the micron to millimeter range can be prepared using laser-induced techniques. In some applications where thicker LIG is required, a 3D printing of LIG on a macroscopic scale has been proposed. Sha et al. [92] prepared complex shape graphene foam by laser induction technique using sucrose and Ni powder as the raw materials. Luong et al. [93] designed a stacked layer object manufacturing method (see Figure 5b). The authors used ethylene glycol to bond two PI films that had formed LIG, and then laser ablated the reverse side of the PI film again. Large volume LIG structures were prepared by repeated bonding and laser treatment. By adjusting the laser frequency, graphene foams with different size resolutions could be obtained.

**Figure 5 nanomaterials-12-02336-f005:**
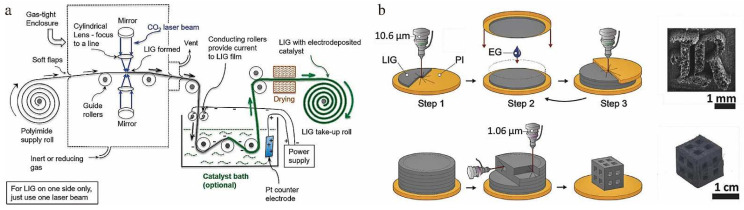
(**a**) Scheme for roll-to-roll production of LIG film, adapted with permission from [90]. (**b**) Schematic of laminated object manufacturing of 3D-printed LIG foams, adapted with permission from [93].

## 4. Applications of LIG

Since the discovery of LIG, many scholars have carried out a large amount of research to improve the preparation process of LIG and apply it to various application fields. This section will introduce the research progress of graphene from the perspective of signal sensing, environmental protection, and energy storage.

### 4.1. Signal Sensing

The excellent physicochemical properties and 3D porous structure of LIG make it an ideal candidate for a variety of sensors. The one-step synthesis of patterned graphene has greatly contributed to the development of smart sensors. Researchers have developed a wide variety of LIG-based sensors or sensor devices with modified LIG materials, such as strain sensors, pressure sensors, temperature sensors, humidity sensors, gas sensors, and biochemical sensors, for monitoring various physical and biochemical signals [94,95,96,97,98,99].

A resistive strain sensor is one of the key components for converting strain stimuli into detectable electrical signals and has great potential in healthcare monitoring, human-machine interfaces, and soft robotics, etc. [100,101,102]. Flexible and stretchable sensors that can be attached to the skin are being developed to monitor the health status of individuals. For example, Luo et al. [103] fabricated a flexible strain sensor with an excellent sensitivity coefficient (GF ≈ 112) by directly generating porous LIG with different patterns on PI films by direct laser writing (DLW). It could identify and monitor different gesture actions and pulses, and could also use finger gestures to control the robotic arm (Figure 6a). Dallinger et al. [104] embedded porous LIG or LIG fibers into 50 μm medical-grade polyurethane, to prepare an LIG-based strain sensor that had excellent stretchability up to 100%. Usually, significant improvements for sensor shape variables are accompanied by a decrease in sensitivity [105,106], therefore, it is a daunting challenge to maintain the balance between sensitivity and strain range of LIG-based strain sensors. Wang et al. [96] investigated a fingerprint-based resistive strain sensor with balanced sensitivity and strain range (Figure 6b). The authors used different process parameters to control the high-temperature gradient in the irradiated region and to rationalize the geometry of the LIG lines (e.g., depth, width, and density). The sensitivity was greatly improved without sacrificing too much inherent flexibility (7.4-fold increase in sensitivity at 42–50% strain, while the strain range was only reduced from 60% to 50%). In addition, this balanced sensitivity and strain sensor could be used to monitor human movement. Take the push-up exercise as an example, a reasonable real-time monitoring of the movement is beneficial for regulating training movements and preventing physical injuries [96]. Zhu et al. [107] prepared a kind of LIG with high conductivity and good mechanical properties on the surface of polydimethylsiloxane (PDMS) films using a diode laser. Then, two identical conductive graphene films were joined with silver paint and copper wire. Finally, they were assembled face-to-face to form a pressure sensor with ultra-high sensitivity (~480 kPa^−1^) and good cycling stability (>4000 repetitive cycles) (Figure 6c). These sensors are considered excellent applications for real-time monitoring of human health.

LIG has high thermal conductivity and low heat capacity, making it an ideal material for thermoacoustic sound sources. Moreover, its porous structure has high sensitivity to weak vibrations and is suitable for sound detection. Therefore, Tao et al. [33] developed a wearable intelligent artificial throat based on LIG, which has both sound generation and detection capabilities (Figure 6d). This intelligent artificial throat detected simple laryngeal vibrations of different intensities or frequencies, such as humming, coughing, and screaming of mute people, and converted them into controllable sounds. LIG-based acoustic source devices have been developed by La et al. [108] and Tao et al. [109] by taking advantage of the nanopore structure, good electrical conductivity and low heat capacity per unit area of LIG, which would be likely used in consumer electronics, multimedia systems, ultrasound detection, and imaging.

Flexible wearable pressure sensors play an important role in advanced applications, such as electronic skin, real-time physiological signal monitoring, and human-computer interaction. Pressure sensors based on different mechanisms have been widely used in past decades [110,111,112,113]. Inspired by bean sprouts, Tian et al. [111] proposed a flexible self-repairing pressure sensor consisting of polystyrene (PS) microspheres as a microspacer core layer sandwiched between two laser-induced graphene/polyurethane (LIG/PU) films (Figure 7a). The porous structure of the LIG provides many cavities for the PS. When subjected to compression, the PS microsphere clusters, which act as spacer layers, modulated the electrical conductance by regulating the degree of physical contact within them. The pressure sensor was highly sensitive, stable, and self-healing. The authors applied it to human arterial pulse monitoring and gait detection, paving the way for scalable production of pressure sensors for human physiological diagnostics and other advanced wearable applications. Li et al. [114] first produced PEEK films with periodic corrugated structures using 3D printing. Porous graphene was then generated on the PEEK films using the LIG technique. Finally, the corrugated LIG (CLIG) films were obtained by transferring them to flexible PDMS films (Figure 7b). This corrugated microstructure facilitated the generation of regular cracks during the stretching process and provided initial line contact under normal compression, which effectively improved the sensor performance. Thus, the authors successfully prepared high-performance strain and pressure sensors based on CLIG films. The CLIG strain sensor had a high resolution of microdeformation (1 μm or 0.01%) and high stability after 15,000 loading cycles. The CLIG pressure sensor had a wide detection range (up to 500 kPa) and high sensitivity (678.2 kPa^−1^). It was reported that the CLIG film can measure wrist pulses, swallowing, and even recognize gestures by the subtle differences in muscle contractions.

Temperature sensors are important tools for real-time temperature monitoring in the fields of healthcare and disease diagnosis [115,116]. LIG technology is of great interest because of its low cost, controllability, and scalability. Due to its high specific surface area, good mechanical stability, electrical and thermal properties, researchers have applied LIG to temperature sensors to improve device performance and reduce preparation costs [117,118]. Kun et al. [119] developed an LIG-based temperature sensor. The sensor was easier to manufacture and operate than conventional thermos-resistance sensors. The accuracy of this sensor was ±0.15 °C, which was better than that of the infrared temperature sensor (±0.30 °C). This LIG-based sensor had an accurate and stable temperature response and could accurately measure the surface temperature of the human body. Recently, Chen et al. [22] developed a fast-response, flexible temperature sensor for non-contact human-machine interface using UV laser RGO (Figure 8a). Experimental results showed that the temperature sensor had the highest sensitivity (0.37% °C^−1^) when the GO concentration was 4 mg/mL and the scan line spacing was 0.12 mm. In addition, this sensor was able to monitor human breathing and contactlessly unlock a combination lock. Han et al. [120] proposed a highly sensitive graphene-based temperature sensor (Figure 8b). It consisted of RGO as a temperature-sensitive layer and LIG as an electrode. The sensor exhibited a high sensitivity of 1.56% °C^−1^ in the range of 25–45 °C. The sensitivity of the RGO/LIG-based temperature sensor decreased with increasing laser power written to the LIG-based electrode.

Due to its three-dimensional porous structure and ultra-high specific surface area, LIG provides sufficient surface locations for gas–solid interactions, making LIG also promising for gas-sensitive detection devices and gas sensing [121,122,123,124]. Stanford et al. [125] proposed a gas sensor based on LIG (Figure 8c). The high surface area and thermal conductivity of LIG ensured fast response times for all studied gases. When different types of gases were inputted, different degrees of resistivity response occurred, due to the difference in thermal conductivity. Gas sensors were also embedded in the cement to form a refractory composite. These sensors were used to determine the composition of various gas mixtures, such as N_2_ and CO_2_, which are the most abundant gas species in flue gas. Thus, LIG-based embeddable sensors can be integrated into composite materials, making electronic functional building materials possible. Yang et al. [126] developed a LIG flexible gas sensing platform with a self-heating function (Figure 8d). This technology used porous graphene as electrodes and highly sensitive nanomaterials (e.g., MoS_2_ and RGO/MoS_2_) as gas-sensitive materials to monitor gases, biomolecules, and chemicals. The platform was composed of a fine wire sensing region of LIG and a serpentine connection region of Ag/LIG. This serpentine design increased the tensile properties of the sensor to accommodate different bending variations of the body. Additionally, the platform had good selectivity at slightly higher self-heating temperatures, which allowed the sensor to detect NO_2_ at a concentration of 1.2 ppb. In addition, Zhang et al. [127] coated a solution mixed with ZnS/SnO_2_ nanoparticles on a PI surface, and CO_2_ laser irradiation was then applied to both sides to convert them into LIG. The LIG, as an electrode, and the semiconductor ZnS/SnO_2_, in the middle, formed an ultraviolet photodetector. The lateral electrode structure reduced the total thickness of the device, thus minimizing strain and improving the flexibility of the photodetector. Due to the high flexibility and the ultra-thin characteristics of graphene, the device showed great mechanical flexibility. This simple and cheap manufacturing process is expected to be applied to the field of miniaturized flexible electronics.

**Figure 8 nanomaterials-12-02336-f008:**
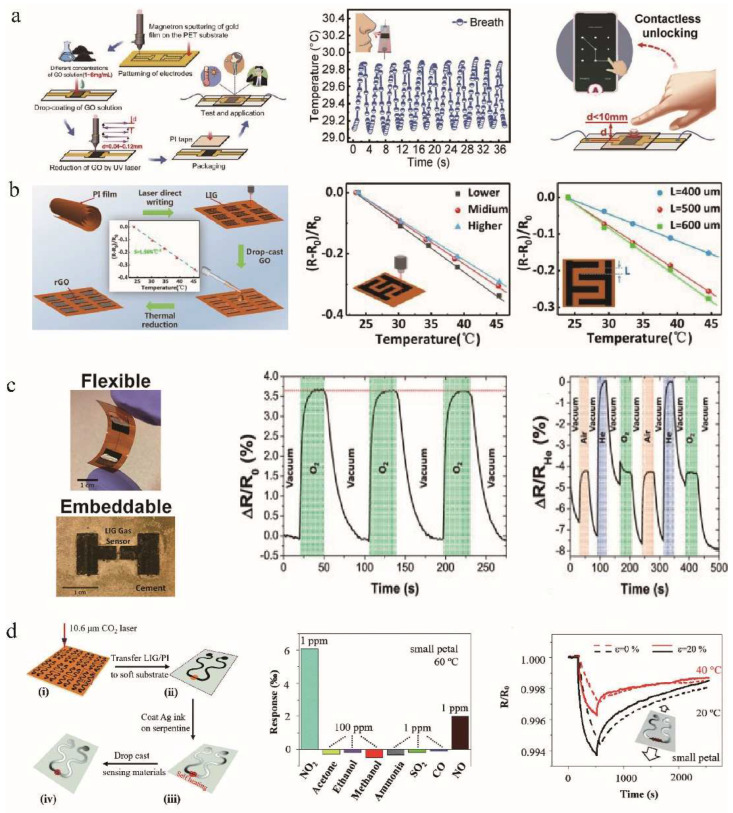
(**a**) Temperature sensor for breathing rate monitoring and contactless combination lock unlocking, adapted with permission from [22]. (**b**) High sensitivity RGO/LIG-based temperature sensor, adapted with permission from [120]. (**c**) LIG-based gas sensor, adapted with permission from [125]. (**d**) Stretchable LIG gas sensing platform. The inset image demonstrates the response of the stretchable gas sensor for NO_2_ of 1 ppm before and after a uniaxial tensile strain of 20% was applied at room temperature and 40 °C, respectively (ppm: parts per million; 1 ppm = 10^−4^%, 1 ppm = 10^3^ ppb), adapted with permission from [126].

With the advantages of masklessness, high resolution, low cost, and high throughput, LIG technology has extended its application to biosensors [25,58]. The conventional screen-printed electrode production process is complicated, time-consuming and expensive, which greatly limits the research progress of electrochemical biosensors [128]. Marques et al. [129] developed a bimolecular system with a double working electrode structure based on LIG technology for the detection of ascorbic acid (AA) and amox icillin (AMOX) (Figure 9a). The combination of electrochemical detection with molecularly imprinted polymer (MIP, a common contaminant in aquaculture) technology could identify specific molecules or compounds, therefore, the two conductive LIG working electrodes modified by MIPs achieved high sensitivity and selectivity for the detection of amoxicillin and ascorbic acid. Cardoso et al. [130] prepared a porous multilayer graphene structure with a resistivity of 102.4 ± 7.3 Ω/square on a PI substrate and used it to design a 3-electrode system (Figure 9b). This system was applied to a biosensor using MIPs as biometric elements. Torrente-Rodríguez et al. [34] reported a fully integrated, flexible, and wireless graphene-based sensor for monitoring the correlation between sweating and circulating cortisol (Figure 9c). This study demonstrated that changes in sweating cortisol could rapidly be determined under acute stress stimuli, revealing the potential of this sensing system to enable dynamic stress monitoring.

From a practical point of view, these above sensors are single-functional and cannot acquire multiple stimuli at the same time. It has been proposed to integrate multiple single-function sensors onto a single substrate, which can detect multiple stimuli simultaneously [131,132]. This fabrication method is expensive and complex. Therefore, a multiparameter sensor that converts each stimulus into an independent signal can overcome the limitations of the above method. Recently, a two-parameter temperature-strain sensor based on a black phosphorus laser-engraved graphene (BP@LEG) heterostructure suitable for electronic skin was investigated by Chhetry et al. [133], (see Figure 9d). The introduced polystyrene-block-poly(ethylene-ran-butylene)-block-polystyrene polymer matrix had excellent mechanical strength, high cycle stability, and good contact with human skin. The thermal index of the hybrid sensor was up to 8106 K in the temperature range of 25–50 °C, the strain sensitivity GF was 2765 (>19.2%), and the detection limit was 0.023%. It had excellent durability in 18,400 cycles. According to the research, this hybrid sensor could be applied to body temperature measurement and the full range of human induced deformation [133].

**Figure 9 nanomaterials-12-02336-f009:**
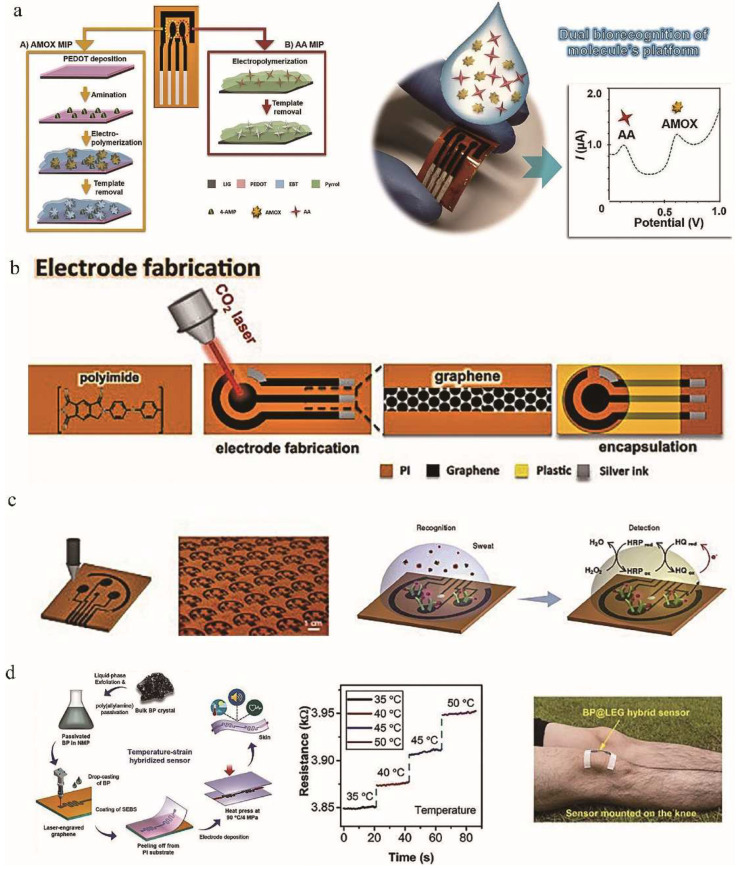
(**a**) Laser-induced graphene-based platforms for dual biorecognition of molecules, adapted with permission from [129]. (**b**) Schematic representation of the workflow employed on the production of the LIG electrodes, adapted with permission from [130]. (**c**) Illustration of the laser-engraving process of a graphene platform, adapted with permission from [34]. (**d**) BP@LEG based hybrid sensor, adapted with permission from [133].

### 4.2. Environmental Protection

Due to its large pore structure, large specific surface area, and excellent surface chemistry, LIG has tremendous potential for many environmental applications, such as anti-pollution systems for desalination and water treatment, air filtration, and generation of anti-bacterial/anti-viral surfaces.

Recently, LIG has been considered as a promising candidate material for air filters due to its good microporous structure and strong adsorption of organic molecules [134]. Stanford et al. [74] investigated a filter with bactericidal effect assembled by LIG (Figure 10a). The device utilized the microporous structure of the LIG membrane to trap bacteria and contaminants, and the periodic Joule heating mechanism of the graphene area to generate localized high temperature of ultra-high 300 °C. It could be efficiently sterilized by high temperature without the effect of biocides, exhibiting the function of self-sterilization and self-cleaning. It is expected to be used in public health and disease control in the future [74].

LIG also has many applications in water treatment due to its large specific surface area, tunable surface properties, excellent antifouling activity, and high photothermal conversion efficiency. The sulfur-doped porous LIG prepared by Singh et al. [78] on sulfone polymer substrates produced surface electrochemical and electrical effects, resulting in highly antimicrobial and excellent antifouling effects. This LIG could be applied to membrane aqueous microfiltration. Polyvinyl alcohol (PVA) is characterized by low toxicity, temperature stability, and high film-forming ability [135]. GO is a two-dimensional nanomaterial that can be used to cover porous or nonporous polymeric membrane supports, resulting in separation membranes with enhanced separation surface properties [136]. Therefore, PVA and GO have widely been used in membrane preparation. Thakur et al. [37,136] loaded PVA and GO onto LIG membranes to prepare LIG-PVA composite membranes and LIG-GO ultrafiltration membranes with controlled performance (Figure 10b,c). These LIG membrane composites had high solute selectivity and permeability comparable to polymeric ultrafiltration membranes. The component of the composite (PVA or GO) could change the surface properties and functionality of the composite membrane, thus determining the degree of antifouling effect. For example, when the amount of graphene oxide was increased, the ultrafiltration membrane increased its rejection of bovine serum albumin to 69%, and antimicrobial resistance from 20 to 99.9%. Under non-filtration conditions, these composite membranes showed 83% less biofilm growth than typical polymeric ultrafiltration membranes, exhibiting excellent antimicrobial properties. These composite membranes had excellent antifouling and antimicrobial properties compared with typical polymeric filtration membranes [136]. Due to the presence of the LIG inner layer, these membranes were also electrically conductive and could effectively purify water by mixed culture of bacteria with applied voltage. Moreover, the presence of PVA greatly improved the mechanical strength of LIG and successfully solved the problem of mechanical strength and insufficient separation performance of conventional LIG-coated membranes. PVA can be used to manufacture highly efficient and environmentally-friendly water purification membranes [37,136].

Since the outbreak of neocoronavirus pneumonia, masks, as filters for both inhaled and exhaled air, have played a critical role in controlling the spread of the epidemic [137]. However, current surgical masks are not self-sterilizing and, therefore, cannot be reused or recycled for other uses. This causes a large amount of medical waste and results in significant economic and environmental stresses. Zhong et al. [75] investigated a medical mask with excellent self-cleaning and photothermal properties (Figure 10d). Since the original medical masks were made of thermoplastics with low melting points (e.g., polypropylene), direct laser transfer of graphene would damage the masks. Therefore, they used a dual-mode laser-induced forward transfer method to deposit a few layers of graphene onto the low-melting-point nonwoven masks. Water droplets were observed to have difficulty remaining on the surface of the treated masks, which had excellent superhydrophobic properties [75]. The hydrophobic LIG was able to effectively inactivate coronaviruses through the synergistic effect of photothermal and hydrophobic properties [138]. After 5 min of sunlight exposure, the surface temperature of the graphene-coated masks rapidly increased to over 80 °C, which is sufficient to extinguish most types of viruses, thus allowing the masks to be reusable after sunlight disinfection. On the other hand, with extension of use, bacteria will accumulate on the mask, and it also becomes a critical issue as to whether the mask can continue to be used. To solve this problem, Huang et al. [36] developed an LIG self-reported antimicrobial mask (Figure 10e) by modulating laser parameters to regulate the surface properties of the LIG, which were used to feed back the safety information of the mask. The wearer's breathing changes the ambient humidity around the device, leading to an inhomogeneous distribution of protons and generating a detectable potential difference or current. Since the accumulation of environmental substances all have a negative impact on the induced potential, this can reflect the number of bacteria or amount of particulate matter that has accumulated on the mask, and thus provide valid information on the suitability of the mask for continued uses. In addition, for evaporation of 10 wt% of brine, the graphene-coated masks exhibited better desalination performance compared with PI solar vaporizers using laser scribing. Therefore, the graphene-coated masks can be directly recycled for solar desalination [75].

Desalination is a key technology to solve the global water scarcity problem. Existing desalination technologies have a high economic cost. Desalination driven by solar energy is a sustainable means of obtaining fresh water. It is important to seek a carbon material with high solar vapor generation efficiency and stable buoyancy. Li et al. [139] reported a floating graphene membrane with high efficiency and scalability for evaporation of seawater into freshwater using entirely solar energy. The PI films were completely converted into graphene films by one-step laser scribing. The LIG film had a solar energy conversion efficiency of 90% under one solar illumination, and evaporated water at a rate of 1.37 kg m^−2^ h^−1^. This high efficiency was due to the efficient water pumping and high optical absorption of the porous structure. Moreover, the authors also desalinated seawater. The experimental results showed that the water with desalination treatment contained fewer electrolytes than the actual seawater or even domestic water. Thus, graphene membranes can indeed be used for seawater desalination. More importantly, these graphene membranes were self-correcting, and floated firmly at the air-water interface, making the process suitable for practical seawater desalination at the ocean surface [139]. 

**Figure 10 nanomaterials-12-02336-f010:**
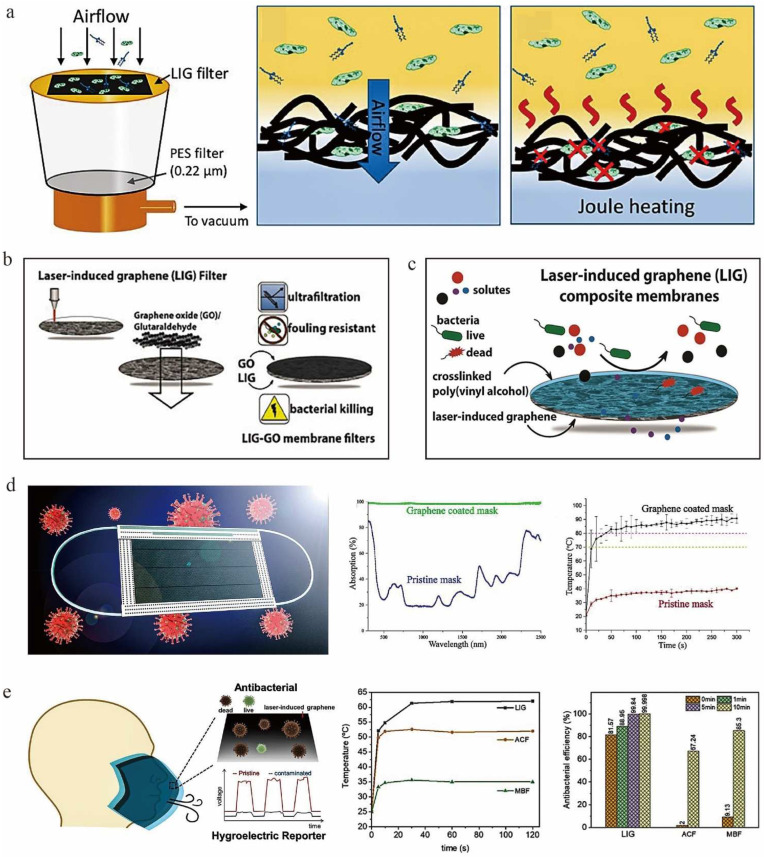
(**a**) Schematic of bacteria capture, sterilization and depyrogenation by Joule-heating, adapted with permission from [74]. (**b**) Preparation of LIG-GO antifouling and sterilizing ultrafiltration membrane, adapted with permission from [136]. (**c**) Schematic diagram of antifouling and bacteriostasis of LIG-PVA composite membrane, adapted with permission from [37]. (**d**) Graphene coated mask for inactivating virus and photothermal performance of pristine and graphene-coated masks, adapted with permission from [75]. (**e**) Self-reporting and photothermally enhanced rapid bacterial killing on a laser-induced graphene mask, adapted with permission from [36].

Recently, Luo et al. [140] prepared a porous LIG film by laser processing on PI@MS film. This LIG film achieved an evaporation rate of 1.31 kg m^−2^ h^−1^ and a photothermal conversion efficiency of 85.4% at 1 solar light intensity. The evaporation performance of the LIG evaporator in high-concentration NaCl solution (10 wt%) could be maintained for up to 12 h, showing excellent stability and salt tolerance (Figure 11a). Inspired by the structure of forests that use sunlight efficiently in nature, Peng et al. [32] obtained an LIG with the target structure, named the forest-like LIG, from polybenzoxazine, using a one-step laser etching process. This prepared LIG film had excellent light absorption and photothermal properties (the average absorption was 99.0% under 1 solar irradiation and the equilibrium temperature was about 90.7 °C). Then, based on this excellent photothermal material, the authors designed a flexible light-driven driver with short response time and high average speed. The authors also designed a salt-resistant bilayer interface solar desalination membrane using the superabsorbent and superhydrophobic properties of the forest-like LIG (Figure 11b). Huang et al. [35] used CO_2_ laser irradiation of PI-coated wood to prepare a porous LIG with pore sizes ranging from hundreds of nanometers to tens of microns at its top and bottom. The middle part of the wood was left intact to maintain microchannels for water transport and low thermal conductivity. This structure was used as a small solar water treatment device (Figure 11c). The device had a superhydrophobic LIG top layer for solar driven desalination and a superhydrophilic LIG bottom layer for effective repulsion of lipophilic organics and antifouling. The LIG technology retained the low thermal conductivity of the wood, and the LIG was a broadband absorbent material with a high evaporation rate.

### 4.3. Energy Storage

Energy storage devices play an important role in energy storage and supply in smart wearable electronics, such as electronic skin and sensors. Due to the excellent electrical conductivity and precise control of the preparation process of LIG, its application has been extended from supercapacitors and micro-supercapacitors to a wide range of energy storage devices, such as lithium metal batteries and fuel cells [24,29,47,84,98,140].

To meet the needs of portable and wearable electronic devices, and modern microelectronic systems, miniature energy storage devices are receiving increasing attention. Micro-supercapacitors (MSCs) are promising energy storage devices with fast response to electrochemical processes, high power density, and cyclic stability [32,35,99,140]. The electrode material of such devices is one of the key factors determining their performance. Lin et al. [25] designed an LIG-MSC in which the LIG was used as the active electrode and current collector. The LIG-MSCs using LIG electrodes had a high specific surface area capacitance and the cyclic voltammograms were pseudo-rectangular at different laser powers, indicating their good double-layer capacitance. The above study further illustrates that LIG is particularly suitable for energy storage devices with good electrochemical properties. In addition, MSCs fabricated with LIG can be stacked to improve the electrochemical performance for commercial applications. Peng et al. [39] extended an approach based on the previous fabrication of MSCs using LIG. Two flexible solid electrolyte supercapacitors, vertically stacked graphene supercapacitors and in-plane graphene micro-supercapacitors, were fabricated by sandwiching the solid polymeric electrolyte PVA and H_2_SO_4_ between two LIG layers (Figure 12a). They both had high electrochemical performance, cyclability, and flexibility. The area capacitance of these devices was up to 9 mF/cm^2^ when the discharge current density was 0.02 mA/cm^2^, which is more than twice as high as when using aqueous solutions [25]. The authors also tested the performance stability of individual LIG-SCs under mechanical bending. The results showed that the performance of bent LIG-SCs was almost the same as that of planar LIG-SCs. This indicated that repeated bending had little effect on the electrochemical performance, further illustrating the unique advantages of 2D LIGs for microcapacitor assembly [39]. To further improve the performance of LIG-SCs, two commonly used strategies are heteroatom doping (e.g., B, N, P, and S) [65] and pseudocapacitive material loading (e.g., Co_3_O_4_ and MnS_2_) [84,105]. The above processes are expensive in raw materials and complex in process, which inevitably limits large-scale economic production. It was shown that LIG could convert electrical energy into thermal energy to perform Joule heating. This Joule heating method was able to reduce the density of defect sites in the graphene structure, which resulted in improved physical and chemical properties [141,142,143]. He et al. [144] introduced Joule heating as a key in situ processing strategy (Figure 12b) combined with laser-induced assembly of graphene paper-based MSCs (LIGP-MSC) to achieve enhanced capacitance.

Although the laser-induced technique has higher fabrication efficiency compared with the conventional method, it is still far from meeting the requirements of rapid mass production and industrialization. Yuan et al. [145] fabricated flexible LIG/MnO_2_ MSCs by the spatially shaped femtosecond laser (SSFL) method. It was reported that the SSFL technique can directly complete the processing of electronic devices in batch. The MSC fabricated by this technology had ultra-high energy density (0.23 Wh cm^−3^), ultra-small time constant (0.01 ms), excellent specific capacitance (128 mF cm^−2^ and 426.7 F cm^−3^), and long-term cycling performance, overcoming the current limitations of low efficiency fabrication and low energy density of micro-supercapacitors. Recently, Le et al. [38] prepared high-quality graphene on fallen leaves using a femtosecond UV laser (Figure 12c). The spatial resolution of the laser pattern was improved due to the ultra-short pulse duration that reduced the thermal diffusion in the heat-affected region. The final obtained femtosecond laser-induced graphene (FsLIG) microelectrodes had excellent electrical conductivity with a resistance of up to 23.3 Ω sq^−1^. Finally, flexible FsLIG-MSCs were also prepared using FsLIG. After 50,000 cycles, the capacitance of the device retained about 99% of the initial capacitance with good electrochemical stability. The maximum surface energy density of the device was 1.204 μWh cm^−2^ and the power density was 324.39 µW cm^−2^, which is comparable to other advanced SCs and MSCs [146,147,148].

With the growth of demand for Li-ion batteries, there is a need to continuously improve the coulomb efficiency, lifetime, stability, and safety factor of batteries. The high porosity, high electrochemical stability, and excellent electronic conductivity of LIG could meet the basic requirements for an ideal collector. A large specific surface area and a good electronic conductivity would reduce the local current density during lithium deposition and slow down the high-volume change of lithium during cycling. Yi et al. [40] laser irradiated PI film on copper foil to obtain an array structure consisting of copper foil, PI column, and LIG (Figure 13a). Among them, the copper foil acted as a conductive channel for lateral electrons, the PI pillar relieved the stress generated by Li deposition, and the LIG film acted as a nucleation site for Li. Moreover, the authors concluded that the large number of defects and heteroatoms in the LIG lowered the nucleation potential barrier of Li, thus promoting stable cycling of the Li-metal anode. As a result, this approach allowed the Li-metal anode to achieve a high coulombic efficiency of 99% at 1 mA cm^−2^ and a high cycle life of 400 h. Yang et al. [149] proposed a novel CuS/Cu_2_S aqueous solution cell. They used a composite of LIG mixed with sulfur as the cathode of this cell. The large specific surface area of LIG sheets (up to 294.69 m^2^ g^−1^) provided a large number of surface-active sites. The characterization revealed that the LIG was multilayered, multi-doped and defective, which facilitated the sulfur loading. Due to the synergistic effect of the new redox couple and LIG, when the current density was 0.8 A g^−1^, the discharge capacity of the battery reached 1654.7 mAh g^−1^ at the initial cycle. Moreover, 91.2% of the reversible capacity was retained after 328 cycles.

Proton exchange membrane fuel cells have received the most attention as fuel cells in recent decades. However, production cost and lifetime limit their commercialization applications [150,151]. Tiliakos et al. [152] transferred non-sacrificial Pt-coated LIG directly onto Nafion membranes as a microporous layer in proton exchange membrane fuel cells by an easily scalable and inexpensive low-temperature decal method (Figure 13b). Due to the more favorable porosity, conductivity, hydrophobicity, and material transport characteristics of LIG, the electrochemical active area of the LIG-based membrane-electrode assembly (MEA) was relatively high at the optimum test conditions of 80 °C and 80% RH. The LIG-based fuel cell effectively improved the power performance by 20%, compared with the reference MEA with the same catalyst loading [152]. Recently, Kong et al. [153] developed an integrated flexible enzyme biofuel cell (EBFC) based on nitrogen-doped graphene obtained by laser scribing method. EBFC is a new type of green energy source and is considered as an alternative power source for wearable devices. Importantly, due to the large specific surface area and good electrical conductivity of laser-scribed N-doped graphene electrodes, electron transfer can be achieved directly without electron mediators. Furthermore, after 20 days of storage, the open-circuit voltage of EBFC still maintained 78% of the initial value, and there was little change after 100 times of bending, indicating that the fuel cell had good stability and mechanical robustness.

**Figure 13 nanomaterials-12-02336-f013:**
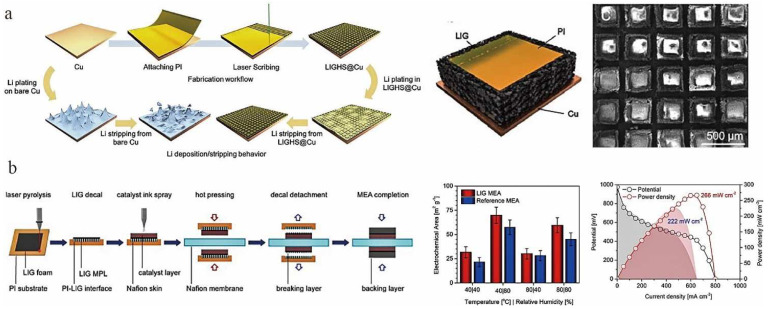
(**a**) Application of LIG in lithium–metal battery, adapted with permission from [40]. (**b**) Application of LIG in proton-exchange membrane fuel cells, adapted with permission from [152].

## 5. Conclusions and Outlook

This review first outlined the preparation methods of graphene, including chemical vapor deposition, redox, mechanical exfoliation, epitaxial growth, and laser-induced technology. Compared with other methods, laser-induced preparation of graphene is highly efficient, low cost, more environmentally friendly, and has a wide selectivity of precursors. Moreover, laser-induced technology can prepare graphene with controllable patterns without using any catalysts or templates. Next, we introduced the research progress of LIG in the preparation process, which has broadened its application fields by adjusting laser parameters, laser type, precursor materials, atmosphere and doping to achieve the morphology control, conductivity, hydrophilic properties, etc. Thirdly, two methods for large-scale production of LIG were briefly discussed. Finally, the applications of LIG in the fields of signal sensing (e.g., various sensors for monitoring human health, medical diagnosis, gas detection, pollutant detection, artificial throats for detection and vocalization), environmental protection (e.g., anti-pollution systems for desalination and water treatment, air filtration, and anti-bacterial/anti-viral surfaces), and energy storage (e.g., supercapacitors, lithium batteries, and fuel cells) were reviewed. 

From the current state of development of LIG, researchers can focus on seeking breakthroughs in the following areas in the development of LIG in the coming years:(1)Although there have been many kinds of graphene preparation technologies, the large-scale, low-cost, environment-friendly, high-quality, and large-size macro-preparation technologies have not yet made substantial breakthroughs, making it difficult to meet the needs of industrial mass production. Exploring new preparation methods, determining more suitable laser parameters, and finding new precursor materials will provide new possibilities for the large-scale production of high-quality graphene;(2)Single-element doping and two-element co-doping have been studied to prepare LIG, which improves the electrochemical properties to a certain extent, and provides a new idea for multi-element co-doping. Therefore, researchers can further explore other new elements and develop multi-element co-doping to endow LIG with more excellent properties, and endow LIG with broader application prospects;(3)Scientists have developed a variety of sensors, such as pressure sensors, strain sensors, temperature sensors, biosensors, gas sensors, etc. From a practical point of view, these sensors have a single function and cannot acquire multiple stimuli at the same time. Studying multifunctional LIG-based sensors that can detect multiple stimuli is a fascinating direction;(4)When LIG is used in masks, its safety is a primary concern. A Canadian research institute has indicated that the use of masks containing graphene may cause the wearer to inhale graphene particles. To the best of the authors' knowledge, there are no relevant reports on the safety assessment of LIG-based masks for this issue so far. Addressing this issue is an important goal for ongoing research.

## Figures and Tables

**Figure 1 nanomaterials-12-02336-f001:**
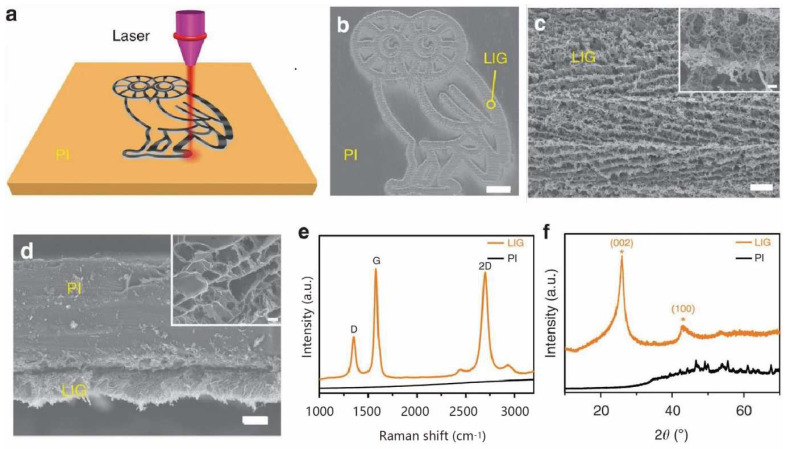
(**a**) Schematic of the synthesis process of LIG from PI. (**b**) Scanning electron microscope (SEM) image of LIG patterned into an owl shape; scale bar, 1 mm. The bright contrast corresponds to LIG surrounded by the darker-colored insulating PI substrates. (**c**) SEM image of the LIG film circled in (**b**); scale bar, 10 μm. Inset is the corresponding higher magnification SEM image; scale bar, 1 μm. (**d**) Cross-sectional SEM image of the LIG film on the PI substrate; scale bar, 20 μm. Inset is the SEM image showing the porous morphology of LIG; scale bar, 1 μm. (**e**) Representative Raman spectrum of an LIG film and the starting PI film. (**f**) XRD of powdered LIG scraped from the PI film, * denotes the peak centroids on the (002) and (100) planes in the LIG structure, adapted with permission from [25].

**Figure 2 nanomaterials-12-02336-f002:**
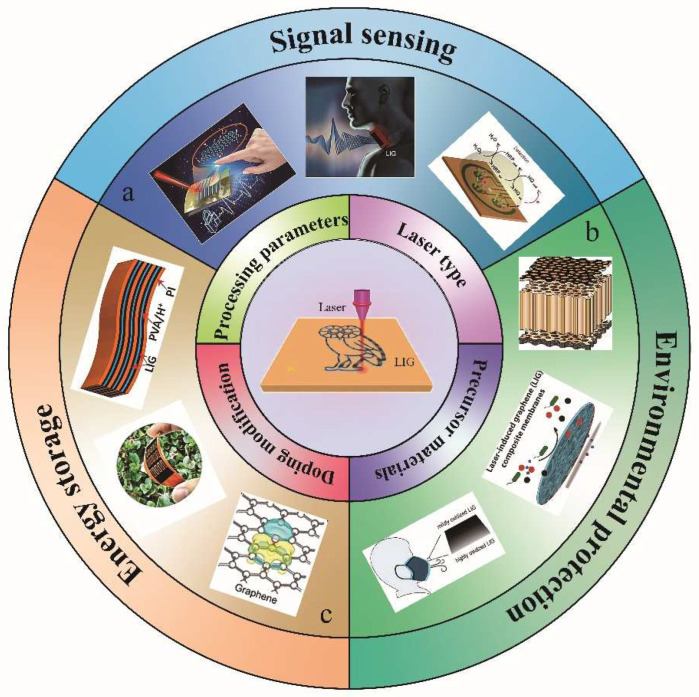
Research progress on the preparation and applications of laser-induced graphene technology. From (**a**) to (**c**) and clockwise: (**a**) Signal sensing, adapted with permission from [22,33,34]. (**b**) Environmental protection, adapted with permission from [35,36,37]. (**c**) Energy storage, adapted with permission from [38,39,40].

**Figure 4 nanomaterials-12-02336-f004:**
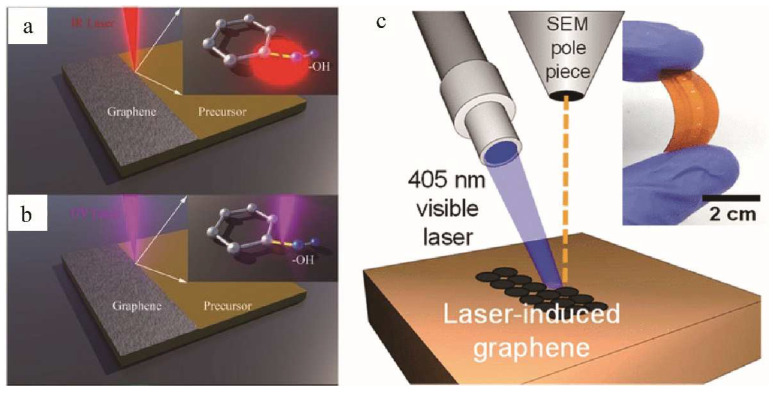
Schematic illustration of (**a**) the photothermal process within the laser writing of graphene electrode; the inset image demonstrates the thermal breaking between C and O–H bonds, adapted with permission from [57]. (**b**) The photochemical process within the laser writing of the graphene electrode; the inset image demonstrates the photon-induced disassociation of band between C and O–H, adapted with permission from [57]. (**c**) Schematic of experimental setup that uses a 405 nm laser mounted on a high-angle port of SEM, adapted with permission from [56].

**Figure 6 nanomaterials-12-02336-f006:**
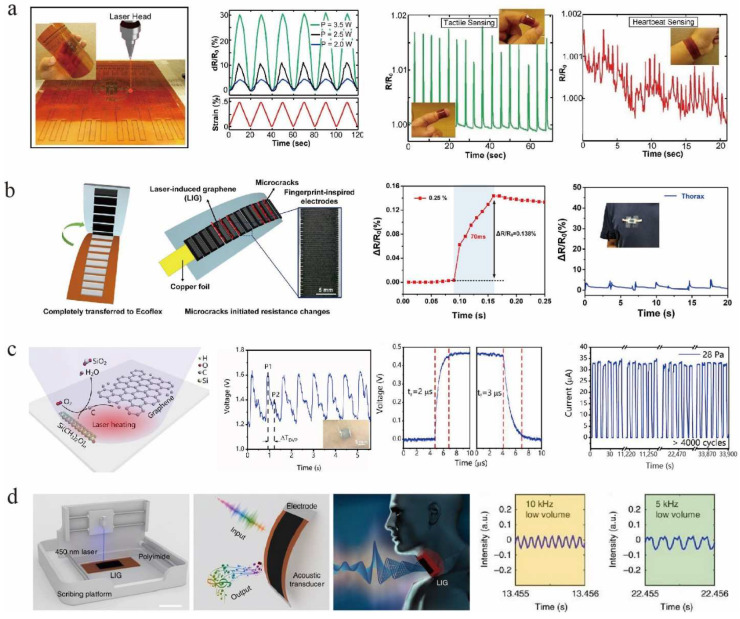
(**a**) Piezoresistivity and applications of the DLW generated graphitic sensors, adapted with permission from [103]. (**b**) Performance of the LIG-based strain sensor inspired by the fingerprint, adapted with permission from [96]. (**c**) Direct laser etching of PDMS films to prepare graphene and characterization of the pressure response, adapted with permission from [107]. (**d**) An artificial LIG-based throat for sound sensing, adapted with permission from [33].

**Figure 7 nanomaterials-12-02336-f007:**
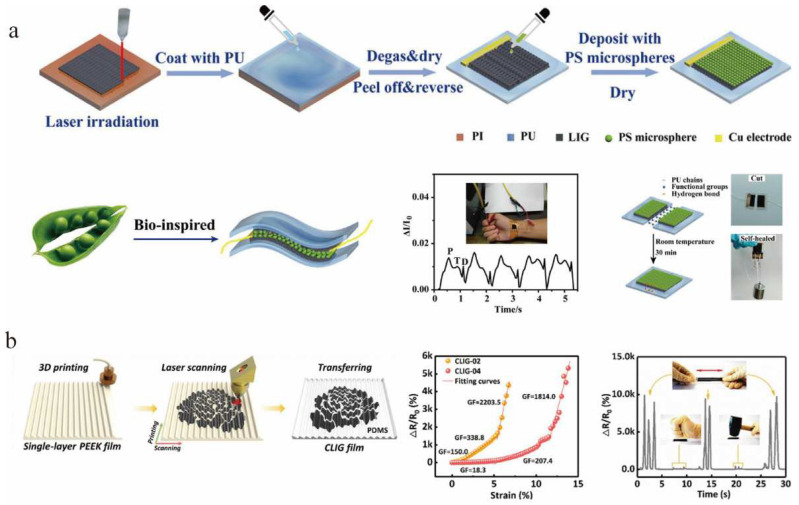
(**a**) Illustration of the bean pod-inspired flexible and healable pressure sensor, adapted with permission from [111]. (**b**) Fabrication process of the CLIG film and characteristics of strain and pressure sensors based on CLIG films, adapted with permission from [114].

**Figure 11 nanomaterials-12-02336-f011:**
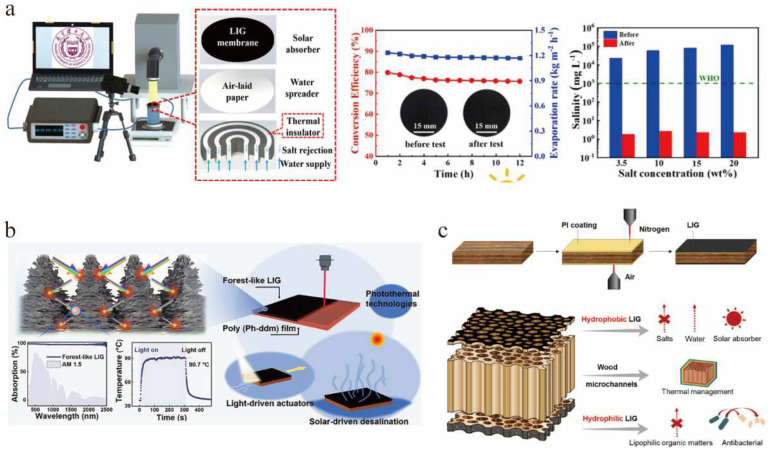
(**a**) Evaporation rate and photothermal conversion efficiency of the LIG-based evaporator, adapted with permission from [140]. (**b**) Forest-like graphene film for light-driven actuators and solar-driven desalination membranes, adapted with permission from [32]. (**c**) hierarchically structured wood-based solar-driven interfacial evaporation system, adapted with permission from [35].

**Figure 12 nanomaterials-12-02336-f012:**
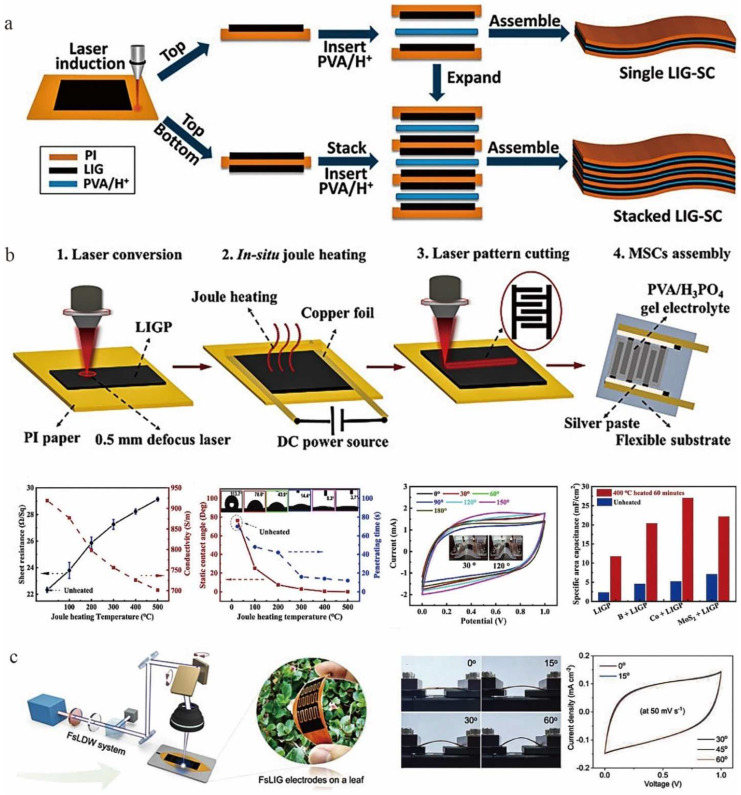
(**a**) Schematic illustration showing the fabrication process for assembling a single LIG-SC and stacked LIG-SC, adapted with permission from [39]. (**b**) The fabrication and performance of J-LIGP-MSCs, adapted with permission from [144]. (**c**) The fabrication and application of FsLIG microelectrodes on fallen leaves, adapted with permission from [38].

**Table 2 nanomaterials-12-02336-t002:** Preparation, properties, and application of doped graphene.

Raw Materials	Doped Materials	Prepared Graphene	Process Method	Performance	Application	Refs.
PI	Metal oxide nanoparticle (Co_3_O_4_, MoO_2_, and Fe_3_O_4_)	MO-LIG	Single irradiation step	Efficient electrocatalytic activity and catalytic stability	Electrocatalyst	[67]
	H_3_BO_3_	B-LIG	Single irradiation step	High energy density, excellent recyclability, and flexibility	Metal-free oxygen reduction reaction catalyst, solar cells, field emission transistors, and lithium ion batteries.	[65]
	Urea	N-LIG	Single irradiation step	Excellent coulombic efficiency, cycling stability, and rate capabilities	Metal ion battery anodes	[86]
	Solid hydrocarbons, elastomers, epoxy, cement, and geopolymer	LIGC	Single irradiation step	Superhydrophobicity, high electrical conductivity	Wearable thermal therapy devices, deicing, anti-icing, antibiofouling and antimicrobial applications	[64]
	Ammonium polyphosphate	P-LIG	Single irradiation step	Good electrochemical performance, and high specific capacitance	Supercapacitor	[87]
	H_3_BO_3_	NB-LIG	Multiple irradiation steps	High peroxidase-like catalytic activity, excellent bactericidal efficiency and capacitive performance	Sterilization, supercapacitor	[29,88]
	Fluorinated ethylene propylene	F-LIG	Single irradiation step	Excellent and stable electrical conductivity and hydrophobicity	Electrode material	[45]
PSU, PES, PPSU	-	S-LIG	Single irradiation step	Electrochemical and antifouling properties	Wastewater purification, fouling-resistant cathodes in microbial fuel cells	[78]

## Data Availability

Not applicable.

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
