# Peer review of "Research Progress on the Preparation and Applications of Laser-Induced Graphene Technology"

_nanomaterials, 2022, doi:10.3390/nano12142336_

Round 1

Reviewer 1 Report

This is an interesting review paper where the authors review laser based preparation methods of graphene and its applications.

I suggest addressing a few revision:

1. Different irradiation condition (namely, by means of cw, ns or fs laser irradiation) will lead to different satial resolution as combination of diffraction limit and underlying light matter interaction.  Spatial resolution of optical patterning with respect to other electronic fib etc…. methods. In section 2, a more detailed analysis of spatial resolution of laser patterning of graphene must be added.

2. The structural quality of LIG (defects, vacancies, doping…) must be discussed with respect to other fabrication methods such as chemical vapor deposition (CVD), mechanical exfoliation. What are the pros and cons of laser induced preparation methods in terms of optical and transport properties.

3. Can the author discuss about large-scale production of LIG?. Other methods such as CVD or liquid phase exfoliation have proven the ability to produce graphene on large scale that is critical for practical application.

Reviewer 2 Report

The current review article demonstrates the current progress in the synthesis and application of laser-processed graphene technology. The manuscript has significant data to be published in this journal. Therefore I recommend Minor revision of this article. My comments are given below -

1. A short note on the effect of laser parameters like wave length, power, pulse width etc. to control the laser treatment should be included in the manuscript.

2. A brief summary of the types of laser treatment should be provided.
